# Molecular mechanism of TRPV2 channel modulation by cannabidiol

Ruth A Pumroy[1], Amrita Samanta[1], Yuhang Liu[2], Taylor ET Hughes[1], Siyuan Zhao[3], Yevgen Yudin[3], Tibor Rohacs[3], Seungil Han[2]*, Vera Y Moiseenkova-Bell[1]*

[1]Department of Systems Pharmacology and Translational Therapeutics, Perelman School of Medicine, University of Pennsylvania, Philadelphia, United States; [2]Pfizer Research and Development, Groton, United States; [3]Department of Pharmacology, Physiology and Neuroscience, New Jersey Medical School, Rutgers University, Newark, United States

**Abstract** Transient receptor potential vanilloid 2 (TRPV2) plays a critical role in neuronal development, cardiac function, immunity, and cancer. Cannabidiol (CBD), the non-psychotropic therapeutically active ingredient of *Cannabis sativa*, is an activator of TRPV2 and also modulates other transient receptor potential (TRP) channels. Here, we determined structures of the full-length rat TRPV2 channel in apo and CBD-bound states in nanodiscs by cryo-electron microscopy. We show that CBD interacts with TRPV2 through a hydrophobic pocket located between S5 and S6 helices of adjacent subunits, which differs from known ligand and lipid binding sites in other TRP channels. CBD-bound TRPV2 structures revealed that the S4-S5 linker plays a critical role in channel gating upon CBD binding. Additionally, nanodiscs permitted us to visualize two distinct TRPV2 apo states in a lipid environment. Together these results provide a foundation to further understand TRPV channel gating, their divergent physiological functions, and to accelerate structure-based drug design.

DOI: https://doi.org/10.7554/eLife.48792.001

*For correspondence:
seungil.han@pfizer.com (SH);
vmb@pennmedicine.upenn.edu (VYM-B)

## Introduction

Transient receptor potential (TRP) channels play significant roles in human physiology and facilitate permeation of essential ions ($Na^+$, $Ca^{2+}$) through the plasma membrane (*Venkatachalam et al., 2014*; *Venkatachalam and Montell, 2007*). Transient receptor potential vanilloid 2 (TRPV2) belongs to the thermoTRPV subfamily of TRP channels (TRPV1-TRPV4), yet TRPV2 does not contribute to thermal nociception in vivo and is insensitive to vanilloids (*Park et al., 2011*; *Perálvarez-Marín et al., 2013*). It has been the least studied among TRPV channels due to the lack of specific pharmacological agonists or antagonists (*Perálvarez-Marín et al., 2013*; *Qin et al., 2008*). Cannabidiol (CBD), a natural product of the *Cannabis sativa* plant, is a TRPV2 agonist (*Qin et al., 2008*) which has been recently used to demonstrate the important role of TRPV2 in the inhibition of glioblastoma multiforme cell proliferation (*Perálvarez-Marín et al., 2013*; *Liberati et al., 2014*; *Nabissi et al., 2010*; *Nabissi et al., 2015*; *Nabissi et al., 2013*). These findings place TRPV2 on the list of important anti-tumor drug targets (*Perálvarez-Marín et al., 2013*; *Liberati et al., 2014*; *Nabissi et al., 2010*; *Nabissi et al., 2015*; *Nabissi et al., 2013*).

Cannabinoids and cannabinoid analogs have been reported to activate a variety of TRP channels, including TRPV1-4 (*Qin et al., 2008*; *De Petrocellis et al., 2012*; *Smart et al., 2000*) and TRPA1 (*De Petrocellis et al., 2008*). CBD activates TRPV2 with an $EC_{50}$ of 3.7 µM for rat TRPV2 (*Qin et al., 2008*), making it a good candidate for the investigation of TRP channel modulation by CBD using cryo-electron microscopy (cryo-EM). Additionally, the activation of other TRP channels by CBD suggests that a cannabinoid binding site could be conserved within this family of channels (*Qin et al.,*

*2008*; *De Petrocellis et al., 2012*; *Smart et al., 2000*; *De Petrocellis et al., 2008*). Understanding the molecular mechanism of CBD modulation of functionally diverse TRP channels could allow us to gain insight into the gating mechanisms of these channels and develop novel therapeutics.

Recently, the TRPV2 channel has been the subject of several cryo-EM studies that offered thought-provoking insights on TRPV2 channel gating (*Huynh et al., 2016*; *Zubcevic et al., 2019a*; *Zubcevic et al., 2016*; *Dosey et al., 2019*). Constructs used in these studies varied from the pore turret deletion mutant (*Zubcevic et al., 2016*) to the full-length engineered resiniferatoxin (RTx)-sensitive channel (*Zubcevic et al., 2019a*) and full-length wild-type channel (*Huynh et al., 2016*; *Dosey et al., 2019*). These studies have been performed in detergent (*Huynh et al., 2016*; *Dosey et al., 2019*), amphipol (*Zubcevic et al., 2019a*; *Zubcevic et al., 2016*), and nanodiscs (*Zubcevic et al., 2019a*) and used TRPV2 from different channel orthologs (*Huynh et al., 2016*; *Zubcevic et al., 2019a*; *Zubcevic et al., 2016*; *Dosey et al., 2019*). Studies using the rabbit TRPV2 pore turret deletion mutant (*Zubcevic et al., 2016*) and full-length engineered resiniferatoxin (RTx)-sensitive channel (*Zubcevic et al., 2019a*) concluded that the pore turret region in rabbit TRPV2 is not essential for channel function and that although mostly unresolved, it adopts a conformation perpendicular to the lipid membrane (*Zubcevic et al., 2019a*; *Zubcevic et al., 2016*). These studies also suggested that engineered RTx-sensitive rabbit TRPV2 channel undergoes symmetry transitions when activated by RTx (*Zubcevic et al., 2019a*; *Zubcevic et al., 2016*). On the other hand, structures using the rat TRPV2 pore turret deletion mutant (*Dosey et al., 2019*) and full-length channel (*Huynh et al., 2016*; *Dosey et al., 2019*) showed that the pore turret region plays a vital role in rat TRPV2 gating, whether in the presence or in the absences of ligands (*Huynh et al., 2016*; *Dosey et al., 2019*). The previous full-length rat TRPV2 structure from our group was determined at ~5 Å resolution in detergent (*Huynh et al., 2016*), but at this resolution we could not build side chains, accurately assign the functional state of the channel or resolve pore turrets (*Huynh et al., 2016*). Recently, the pore turret structure in the full-length rat TRPV2 channel in detergent has been resolved (*Dosey et al., 2019*) and shown to adopt a conformation parallel to the lipid membrane. Moreover, this full-length rat TRPV2 structure in detergent has been assigned to the open state of the channel (*Dosey et al., 2019*), however the provided cryo-EM data were not of sufficient quality to independently confirm this proposed channel conformation or the pore turret structure of the channel.

Here, we reconstituted the full-length rat TRPV2 channel in nanodiscs and determined the effect that a lipid environment and 30 µM CBD have on the channel structure. We now report two TRPV2 structures determined in an apo state resolved to 3.7 Å (TRPV2$_{APO\_1}$) and 4.0 Å (TRPV2$_{APO\_2}$) and two TRPV2 structures determined in the presence of 30 µM CBD resolved to 3.4 Å (TRPV2$_{CBD\_1}$) and 3.2 Å (TRPV2$_{CBD\_2}$) (*Figure 1*). In both TRPV2$_{CBD\_1}$ and TRPV2$_{CBD\_2}$, CBD density occupies the newly proposed CBD-binding site between S5 and S6 helices of adjacent subunits, but both channels adopt non-conducting conformations (*Figure 1*). While TRPV2$_{CBD\_1}$ is very similar to TRPV2$_{APO\_1}$, TRPV2$_{CBD\_2}$ diverges from the non-conducting TRPV2$_{APO\_1}$ state in nanodiscs, suggesting that we have resolved the channel in CBD-bound pre-open states or desensitized states. Moreover, nanodiscs allow us to observe TRPV2$_{APO\_2}$ state in a partially open conformation, suggesting that lipids play an important role in channel transitions between closed and open states. Comparison between the newly determined TRPV2 structures also revealed that the S4-S5 linker plays a critical role in channel gating. Together these results provide new structural information on the TRPV2 channel which could be used for the design of novel TRP channel therapeutics.

## Results

To improve on previous low resolution apo full-length rat TRPV2 structures (*Huynh et al., 2016*) and identify the CBD binding site, we reconstituted full-length rat TRPV2 into nanodiscs. To capture TRPV2 in the CBD-bound state, we incubated nanodisc-reconstituted TRPV2 for 30 min with 30 µM CBD before preparing cryo grids. This corresponds to ten times the reported EC$_{50}$ of CBD for TRPV2 (*Qin et al., 2008*) and is comparable to reported ligand concentrations and incubation time for cryo-EM studies with truncated TRPV1 (*Cao et al., 2013*).

All cryo-EM data obtained for apo and 30 µM CBD-bound TRPV2 reconstituted into nanodiscs was analyzed using RELION 3.0 without symmetry imposition during the initial reconstruction and classification (*Zivanov et al., 2018*; *Scheres, 2016*; *Scheres, 2012*). After the initial round of 3D

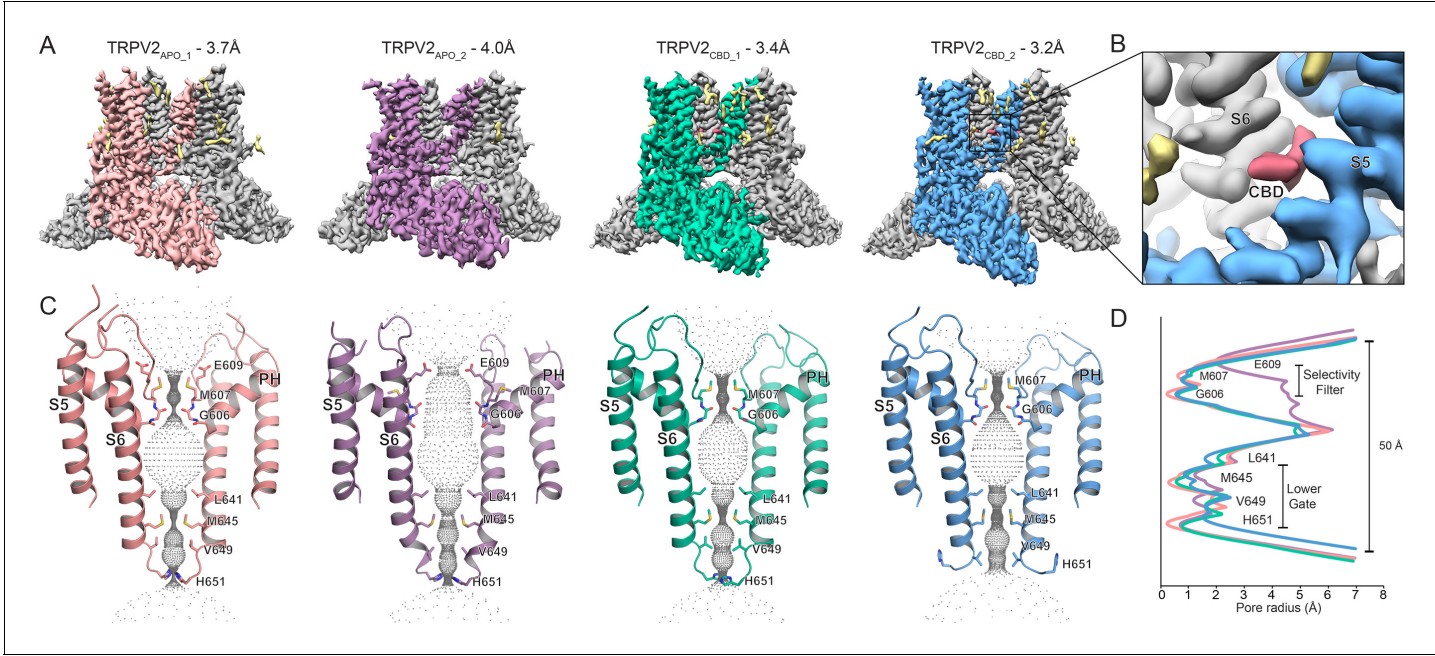

**Figure 1.** Overview of the full-length rat TRPV2 channel structures in nanodiscs. (**A**) Three-dimensional cryo-EM reconstructions for TRPV2_APO_1 at 3.7 Å (salmon), TRPV2_APO_2 at 4.0 Å (purple), TRPV2_CBD_1 at 3.4 Å (green), TRPV2_CBD_2 at 3.2 Å (blue); (**B**) CBD binding pocket in the TRPV2_CBD_2 structure. The S5 and S6 helices are shown in blue and gray, respectively. Density attributed to CBD is shown in pink. (**C**) The profile of the ion permeation pathway for TRPV2_APO_1 (salmon), TRPV2_APO_2 (purple), TRPV2_CBD_1 (green) and TRPV2_CBD_2 (blue) structures; (**D**) Graphical representation of the radius of the pore as a function of the distance along the ion conduction pathway.

DOI: https://doi.org/10.7554/eLife.48792.002

The following figure supplements are available for figure 1:

**Figure supplement 1.** EM summary of apo full-length rat TRPV2 in nanodiscs.
DOI: https://doi.org/10.7554/eLife.48792.003

**Figure supplement 2.** Apo TRPV2 cryo-EM data processing.
DOI: https://doi.org/10.7554/eLife.48792.004

**Figure supplement 3.** EM summary of CBD-bound full-length rat TRPV2 in nanodiscs.
DOI: https://doi.org/10.7554/eLife.48792.005

**Figure supplement 4.** CBD-bound TRPV2 cryo-EM data processing.
DOI: https://doi.org/10.7554/eLife.48792.006

**Figure supplement 5.** Representative densities from the TRPV2_APO_1 cryo-EM map.
DOI: https://doi.org/10.7554/eLife.48792.007

**Figure supplement 6.** Representative densities from the TRPV2_APO_2 cryo-EM map.
DOI: https://doi.org/10.7554/eLife.48792.008

**Figure supplement 7.** Representative densities from the TRPV2_CBD_1 cryo-EM map.
DOI: https://doi.org/10.7554/eLife.48792.009

**Figure supplement 8.** Representative densities from the TRPV2_CBD_2 cryo-EM map.
DOI: https://doi.org/10.7554/eLife.48792.010

**Figure supplement 9.** Apo full-length rat TRPV2 pore turret comparison.
DOI: https://doi.org/10.7554/eLife.48792.011

**Figure supplement 10.** Cryo-EM density in the CBD binding pocket.
DOI: https://doi.org/10.7554/eLife.48792.012

classification, the best set of particles from each dataset was evaluated by the Map Symmetry function in PHENIX (*Adams et al., 2002*) and assigned four-fold symmetry based on the highest correlation coefficient score (*Figure 1*, *Figure 1—figure supplements 1–4*, *Table 1*).

The apo TRPV2 data yielded two structures: TRPV2_APO_1 at 3.7 Å and TRPV2_APO_2 at 4.0 Å (*Figure 1*, *Figure 1—figure supplements 1–2*, *Table 1*). The CBD-bound TRPV2 data also yielded two structures: TRPV2_CBD_1 at 3.4 Å and TRPV2_CBD_2 at 3.2 Å (*Figure 1*, *Figure 1—figure supplements*

**Table 1.** Cryo-EM data collection and model statistics.

| | TRPV2$_{APO\_1}$ (EMD-20677, PDB 6U84) | TRPV2$_{APO\_2}$ (EMD-20678, PDB 6U85) | TRPV2$_{CBD\_1}$ (EMD-20686, PDB 6U8A) | TRPV2$_{CBD\_2}$ (EMB-20682, PDB 6U88) |
|---|---|---|---|---|
| **Data collection and processing** | | | | |
| Magnification | 81,000x | | 81,000x | |
| Detector mode | super-resolution | | counting | |
| Voltage (kV) | 300 | | 300 | |
| Defocus range (μm) | 0.8–3.0 | | 0.8–3.0 | |
| Pixel size (Å) | 1.06 | | 1.06 | |
| Total extracted particles (no.) | 1,181,347 | | 2,290,820 | |
| Refined particles (no.) | 598,859 | | 710,728 | |
| Final particles (no.) | 42,407 | 14,332 | 23,944 | 36,153 |
| Symmetry imposed | C4 | C4 | C4 | C4 |
| Map sharpening $B$ factor (Å$^2$) | −82 | −88 | −51 | −38 |
| Map resolution (Å) FSC threshold | 3.7 0.143 | 4.0 0.143 | 3.4 0.143 | 3.2 0.143 |
| **Model Refinement** | | | | |
| Model resolution cut-off (Å) | 3.7 | 4.0 | 3.4 | 3.2 |
| Model composition Nonhydrogen atoms Protein residues Ligands | 20,460 2520 0 | 20,140 2472 0 | 20,036 2468 CBD: 4 | 20,016 2444 CBD: 4 |
| R.M.S. deviations Bond lengths (Å) Bond angles (°) | 0.011 0.742 | 0.007 0.664 | 0.008 0.947 | 0.009 1.060 |
| Validation MolProbity score Clashscore Poor rotamers (%) CαBLAM outliers (%) EMRinger score | 1.69 5.36 0.90 1.14 1.72 | 2.06 11.75 0.18 0.84 3.03 | 1.37 3.39 0.37 0.50 2.61 | 1.38 4.23 0.55 2.18 2.55 |
| Ramachandran plot Favored (%) Allowed (%) Disallowed (%) | 93.97 6.03 0.00 | 92.31 7.69 0.00 | 96.38 3.62 0.00 | 96.97 3.03 0.00 |

DOI: https://doi.org/10.7554/eLife.48792.013

3–4, *Table 1*). The good quality of these cryo-EM maps allowed us to build atomic models for the TRPV2 channel in four different states, placing sidechains throughout the models. (*Figure 1*, *Figure 1—figure supplements 1–8*, *Table 1*). As in previous cryo-EM structures of TRPV2 (*Zubcevic et al., 2016*; *Huynh et al., 2016*; *Dosey et al., 2019*; *Zubcevic et al., 2019a*), all four newly resolved full-length rat TRPV2 channels form homo-tetramers featuring six transmembrane helices (S1-S6) spanning the transmembrane domain (TMD) with six ankyrin repeat domains (ARDs) splayed out like a pinwheel on the cytoplasmic face of the protein, with the ARDs of adjacent monomers connected through a β-sheet region composed of the N-linker between the ARDs and the TMD and a portion of the C-terminus. S1-S4 form a bundle, while S5, S6 and the pore helix (PH) extend away from S1-S4 to domain swap with adjacent monomers and form the pore. Despite working with the full-length rat TRPV2 protein, all four maps obtained in this study lack fully resolved density for the pore turret between the top of S5 and the pore helix (*Figure 1*, *Figure 1—figure supplement 9*). Nevertheless, partial density for this region of the channel indicates that the pore turret forms a flexible loop perpendicular to the lipid membrane, which agrees with observations in the full-length engineered RTx-sensitive rabbit TRPV2 channel (*Zubcevic et al., 2019a*). Thus, our data do not agree with the observation that the charged and polar flexible pore turret loop is

arranged parallel to the plasma membrane as it has been reported for the full-length rat TRPV2 structure in detergent (*Dosey et al., 2019*) (*Figure 1*, *Figure 1—figure supplement 9*).

Comparison between full-length rat TRPV2 in the apo (TRPV2$_{APO\_1}$, TRPV2$_{APO\_2}$) and the CBD-bound (TRPV2$_{CBD\_1}$, TRPV2$_{CBD\_2}$) states in nanodiscs allowed us to identify the CBD-binding pocket, which is located between the S5 and S6 helices of adjacent TRPV2 monomers (*Figure 1A–B*). The non-protein density assigned to the CBD ligand is not present in the final or half maps of either apo TRPV2 structure and is present in all half maps of the CBD-bound structures (*Figure 1A–B*, *Figure 1—figure supplement 10*). Since this density is unique to the dataset with added CBD and is similar in size and shape to CBD, we have assigned it to CBD (*Figure 1A–B*, *Figure 2—figure supplement 1*).

All full-length rat TRPV2 maps presented in this manuscript also have extra non-protein density that could be attributed to lipids (*Figure 1A*). Several cryo-EM studies on TRPV channels also observed very similar lipid densities in different TMD hydrophobic pockets and proposed that some of these lipids are important for channel function and structure (*Gao et al., 2016*; *Hughes et al., 2018a*; *McGoldrick et al., 2018*; *Singh et al., 2018*; *Zubcevic et al., 2018a*). It is interesting to note that ligand binding sites in several other TRPV channel cryo-EM structures determined so far have been also occupied by lipids in their apo structures (*Gao et al., 2016*; *Hughes et al., 2018a*; *McGoldrick et al., 2018*; *Singh et al., 2018*; *Zubcevic et al., 2018a*), which is not the case for the CBD binding pocket determined here (*Figure 1A*).

The pore analysis of these four structures showed that they adopt distinct non-conducting conformations (*Figure 1C–D*). The TRPV2$_{APO\_1}$ pore is occluded at the selectivity filter (Gly606 and Met607) and the lower gate of the channel (Met645 and His651), representing a closed TRPV2 state in nanodiscs. On the other hand, the TRPV2$_{APO\_2}$ pore selectivity filter is wide open, suggesting that this structure is in a partially open state. The TRPV2$_{CBD\_1}$ and TRPV2$_{CBD\_2}$ pores are in two different non-conducting states, suggesting that we may have resolved CBD-bound pre-open or desensitized states of the channel.

## The CBD binding site

A comparison of the models built into the TRPV2$_{APO\_1}$, TRPV2$_{CBD\_1}$, and TRPV2$_{CBD\_2}$ cryo-EM maps revealed that the CBD binding site is lined with mostly hydrophobic and aromatic residues. These include Leu631, Tyr634, Val635 and Leu638 on the S6 helix of one monomer and Leu537, Phe540, Leu541 on the S5 helix and Met640 on the S6 helix of an adjacent monomer, while the pore helix of the adjacent monomer caps the pocket with residues Phe601 and Thr604 (*Figure 2*). CBD is composed of a cyclohexene headgroup with small hydrophobic substitutions, a middle aromatic ring with two free phenolic hydroxyl groups, and a 5-carbon acyl tail (*Figure 2—figure supplement 1*). The hydrophobic head group enters furthest into the CBD-binding pocket, where it is surrounded by aromatic and hydrophobic residues. Notably, although most of the residues in this pocket are in similar positions in all three structures, Tyr634 rotates between TRPV2$_{APO\_1}$ and the CBD-bound structures (*Figure 2*, *Figure 2—figure supplement 2*) to sit next to CBD, which may be due to the hydrophobic effect shielding CBD from ions in the pore. The two free phenolic hydroxyl groups of the middle region of CBD fit between turns of the α-helices on either side, potentially coordinating via bifurcated hydrogen bonds with the carbonyl of Leu631 and amide of Tyr634 on S6 of one monomer and the carbonyl of Leu537 and the amide of Leu541 on S5 of the adjacent monomer (*Figure 2—figure supplement 2*). Additionally, the Leu537 sidechain has rotated in the CBD-bound structures to accommodate the aromatic ring of CBD (*Figure 2*). Density for the CBD acyl tail trails off (*Figure 2—figure supplement 1*), indicating flexibility where the acyl tail meets lipids in the lipid bilayer.

CBD was oriented with the headgroup on the interior of the pocket and the tail extending into the lipid bilayer because the ligand density on the interior of the pocket was a poor fit for a linear acyl tail and this configuration did not stay stably in the density during PHENIX refinements. An alternate pose of the ligand was also tested, with CBD rotated 180° around its longest axis (*Figure 2—figure supplement 1*), turning the headgroup over in the interior of the pocket. While this pose was stable in the ligand density during refinement, it clashed with the Leu537 sidechain in the TRPV2$_{CBD\_1}$ structure, indicating the original pose as the optimal fit for this data (*Figure 2—figure supplement 1*).

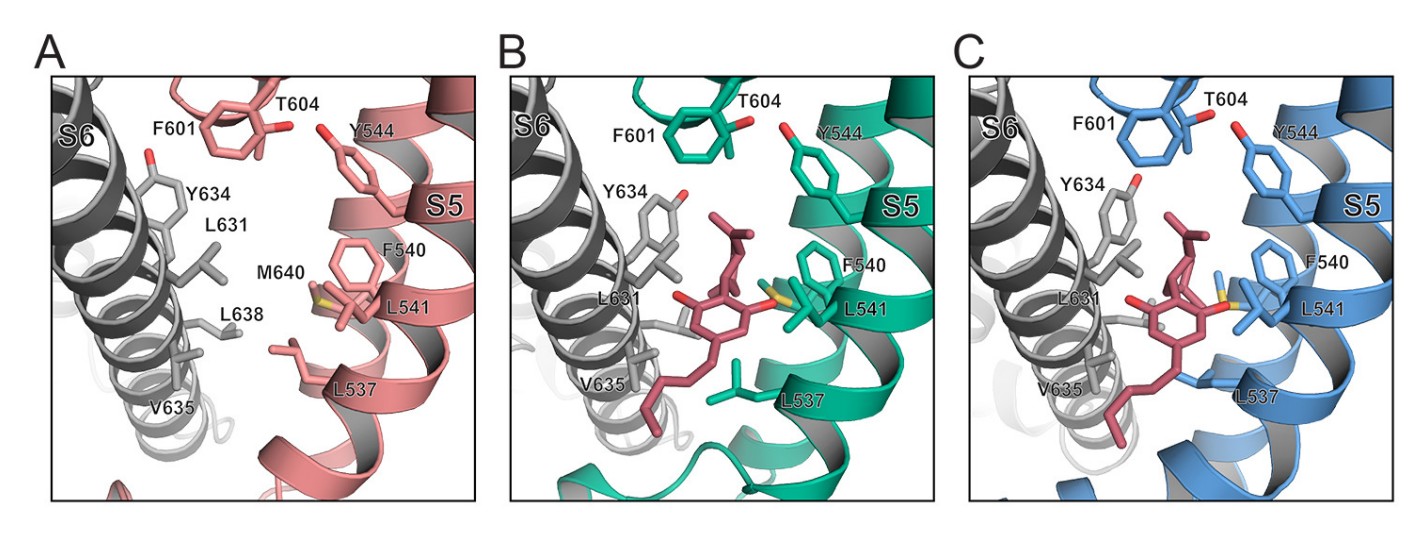

**Figure 2.** The CBD binding site. Model representations of the CBD binding pockets in the (A) TRPV2$_{APO\_1}$ (salmon), (B) TRPV2$_{CBD\_1}$ (green) and (C) TRPV2$_{CBD\_2}$ (blue) structures. CBD is shown as pink sticks. Residues of interest are labeled and represented as sticks.

DOI: https://doi.org/10.7554/eLife.48792.014

The following figure supplements are available for figure 2:

**Figure supplement 1.** CBD fit in the cryo-EM map.
DOI: https://doi.org/10.7554/eLife.48792.015
**Figure supplement 2.** Hydrogen binding between CBD and the helical backbone.
DOI: https://doi.org/10.7554/eLife.48792.016
**Figure supplement 3.** Sequence alignment of thermoTRPV channels.
DOI: https://doi.org/10.7554/eLife.48792.017
**Figure supplement 4.** The double mutant L541F-L631F increases current responses to CBD.
DOI: https://doi.org/10.7554/eLife.48792.018

Based on the CBD binding site identified in out cryo-EM maps, we predicted that mutating some of the smaller hydrophobic residues in this pocket to a bulky aromatic residue like phenylalanine could alter CBD access to the pocket (*Figure 2*). Many of the residues on the interior of the binding pocket, which make contacts with the hydrophobic head group are already bulky aromatic groups and are additionally highly conserved across thermoTRPV channels, possibly indicating a crucial role at the tetramerization interface (*Figure 2—figure supplement 3*), so we focused on mutating Leu541, Leu631, and Val635 (*Figure 2*). Whole cell patch clamp recordings showed that the ratio of currents induced by 20 µM CBD and 100 µM 2-APB was higher in cells transfected with the Leu541-Phe-Leu631Phe mutant compared to wild type TRPV2 (*Figure 2—figure supplement 4*). Neither 20 µM CBD nor 100 µM 2-APB induced any current in non-transfected cells and in cells transfected with the Val635Phe mutant, indicating that the latter mutant was non-functional (data not shown). These electrophysiological results suggest that residues Leu541 and Leu631 may be involved in CBD modulation of the channel.

## Structural changes on CBD binding

To assess the effect CBD binding had on the TRPV2 structure, TRPV2$_{APO\_1}$ was aligned with the TRPV2$_{CBD\_1}$ and TRPV2$_{CBD\_2}$ structures. The alignment of these three structures based on the tetrameric pore (between residues Asp536-His651) had a low root mean square deviation (R.M.S.D.): 0.4 Å for TRPV2$_{APO\_1}$ to TRPV2$_{CBD\_1}$, and 0.5 Å for TRPV2$_{APO\_1}$ to TRPV2$_{CBD\_2}$. While the global alignment of the TRPV2$_{APO\_1}$ with TRPV2$_{CBD\_1}$ had an R.M.S.D. of 0.6 Å, the global alignment of TRPV2$_{APO\_1}$ with TRPV2$_{CBD\_2}$ had a higher R.M.S.D. of 1.9 Å.

Based on the tetrameric pore alignment, TRPV2$_{APO\_1}$ and TRPV2$_{CBD\_1}$ are essentially identical structures (*Figure 3A–C*, *Figure 3—figure supplement 1*). The cryo-EM map quality for these structures allowed us to resolve the distal C-terminus for both of these structures up to residue Leu719

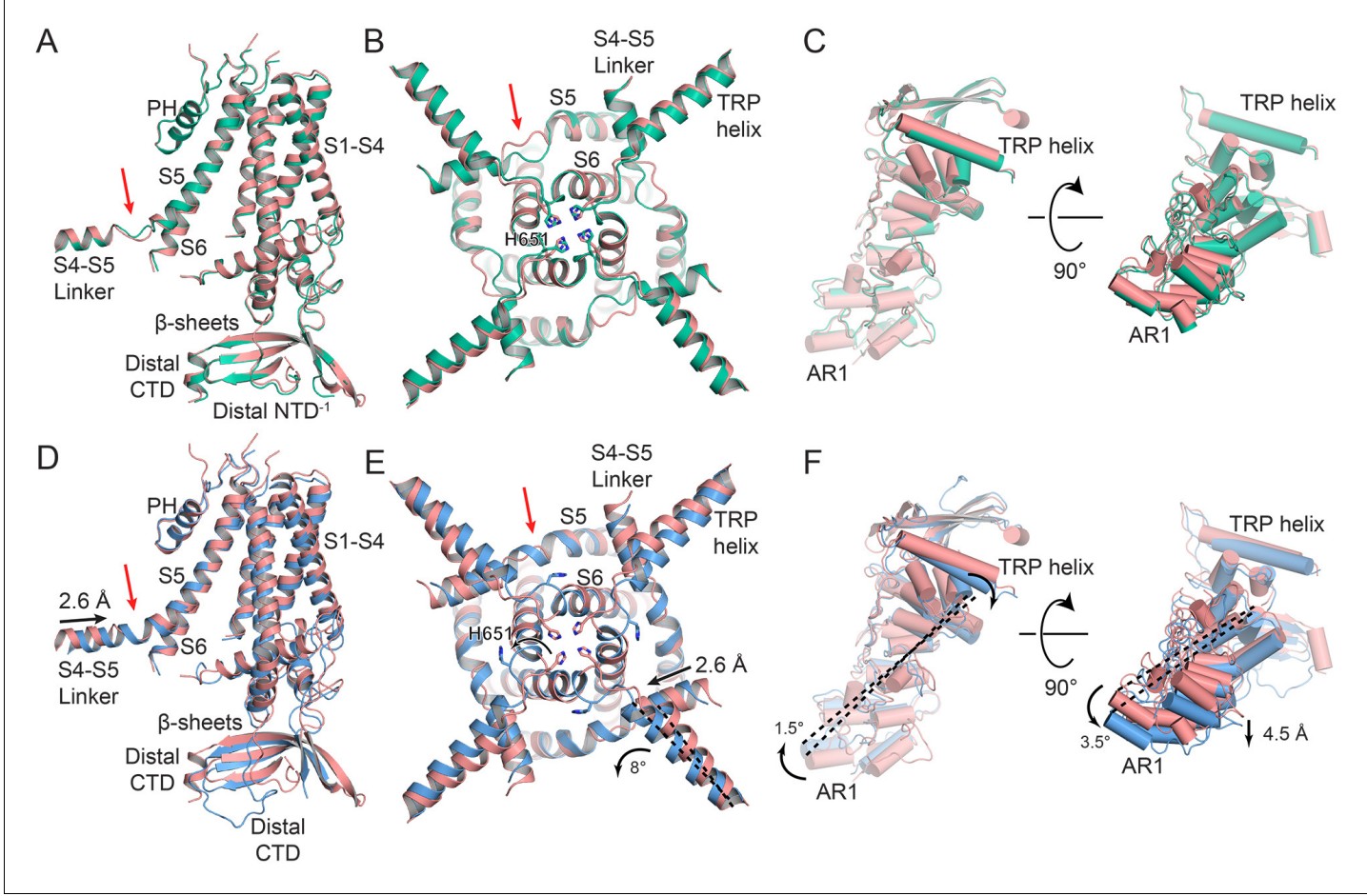

**Figure 3.** Conformational changes upon CBD binding. (**A**) Overlay of the TRPV2$_{APO\_1}$ (salmon) and TRPV2$_{CBD\_1}$ (green) structures, aligned to the tetrameric pore (S5–PH–S6). (**B**) Zoom view from the intracellular side of the membrane on the overlay between TRPV2$_{APO\_1}$ (salmon) and TRPV2$_{CBD\_1}$ (green) structures. (**C**) Overlay of one ARD from the TRPV2$_{APO\_1}$ (salmon) and TRPV2$_{CBD\_1}$ (green) structures with helices displayed as cylinders. (**D**) Overlay of the TRPV2$_{APO\_1}$ (salmon) and TRPV2$_{CBD\_2}$ (blue) structures, aligned to the tetrameric pore (S5–PH–S6). (**E**) Zoom view from the intracellular side of the membrane on the overlay between TRPV2$_{APO\_1}$ (salmon) and TRPV2$_{CBD\_2}$ (blue) structures. (**F**) Overlay of one ARD in the TRPV2$_{APO\_1}$ (salmon) and TRPV2$_{CBD\_2}$ (blue) structures with helices displayed as cylinders. Residues of interest are represented as sticks and labeled. Red arrows indicate the junction between the S4-S5 linker and S5. Dashed black lines are references to indicate rotation. The measurements for rotation and lateral shifts are labeled and indicated with black arrows.

DOI: https://doi.org/10.7554/eLife.48792.019

The following figure supplements are available for figure 3:

**Figure supplement 1.** Cryo-EM map density around the S4-S5 linker, S5, and the TRP helices.

DOI: https://doi.org/10.7554/eLife.48792.020

**Figure supplement 2.** Cryo-EM maps highlighting the CTD and NTD.

DOI: https://doi.org/10.7554/eLife.48792.021

**Figure supplement 3.** Cryo-EM map density around the CTD and NTD of TRPV2$_{APO\_1}$ and TRPV2$_{CBD\_2}$.

DOI: https://doi.org/10.7554/eLife.48792.022

(*Figure 3A*, *Figure 3—figure supplements 2–3*). The distal C-terminus formed a short helix that interacts with the N-terminal β-sheet region and ARD of the adjacent subunit of the channel (*Figure 3—figure supplement 3*). For both these structures, Trp715 forms CH-π interactions with Pro337, found in the N-linker β-sheet region, while Glu716 forms polar interactions with Asn263, found in AR5 of the adjacent subunit of the channel (*Figure 3—figure supplement 3*). Moreover, in both of these structures we resolved a distal N-terminus that can be fit to residues Gln30-Asn45 and which makes contacts between the ARDs of one monomer and the β-sheet region of the adjacent monomer (*Figure 3—figure supplements 2–3*). Met35 lies in a hydrophobic and aromatic pocket

formed by the loops between AR3, AR4 and AR5 and the β-sheet region of an adjacent monomer and makes contacts with Trp333. Phe39 sits in a hydrophobic pocket formed by Phe330, Leu 342, Leu 689 and Val681 in the β-sheet region of the adjacent monomer. Lys123 forms as salt bridge with Glu36 and Lys118 interacts with the backbone carbonyl of Met35 (*Figure 3—figure supplement 3*). The difference between these two structures is a minor shift occurring at the S4-S5 linker, which is likely necessary to accommodate the adjacent CBD molecule, so TRPV2$_{CBD\_1}$ remains in a CBD-bound but non-conductive state (*Figure 3A–B*).

Alignment between TRPV2$_{APO\_1}$ and TRPV2$_{CBD\_2}$ revealed several deviations between these two structures (*Figure 3D–F*). In TRPV2$_{APO\_1}$ and TRPV2$_{CBD\_1}$ there is a break between the helices of S5 and the S4-S5 linker (*Figure 3A*, *Figure 3—figure supplement 1*), while in TRPV2$_{CBD\_2}$ the S4-S5 linker is 2.6 Å closer to the S5 helix, bringing the two helices closer together for the formation of a continuous helix (*Figure 3D–E*, *Figure 3—figure supplement 1*). The quality of density in this region is not sufficient to determine whether this caused the formation of the S4-S5 π helix hinge (*Figure 3—figure supplement 1*), which has been observed in the S4-S5 linker in the full-length engineered RTx-sensitive rabbit TRPV2 channel (*Zubcevic et al., 2019a*). The conformational change in the S4-S5 linker of TRPV2$_{CBD\_2}$ is coupled to an 8° rotation of the TRP helix (*Figure 3E*, *Figure 3—figure supplement 1*). This rotation of the TRP helix in TRPV2$_{CBD\_2}$ also shifted it closer to the bottom of S6, potentially allowing for the rotation of His651 away from the pore of the channel (*Figure 3E*). The rotation of the TRP helix in TRPV2$_{CBD\_2}$ is also paired with a pivot outward by 1.5° and downward by 3.5° of each individual ARD, resulting in a downward swing of 4.5 Å at ankyrin repeat (AR) 1 (*Figure 3F*). Moreover, the distal C-terminus in TRPV2$_{CBD\_2}$ transitioned from a helix to a coil, extending to residue Pro729 that wrapped around the β-sheet region and made contacts with AR 2–3 of the adjacent subunit of the channel (*Figure 3D*, *Figure 3—figure supplement 2*). In TRPV2$_{CBD\_2}$, the Trp715 is out of range of Pro337, but instead is able to form a cation-π interaction with Arg175, found in AR3. Glu716 rotates away and makes no contacts with the ARDs or β-sheet region (*Figure 3B–C*, *Figure 3—figure supplement 3*). Pro726 lies in a hydrophobic and aromatic pocket formed by the loops between AR3, AR4 and AR5 and the β-sheet region of an adjacent monomer and makes contacts with Trp333. Leu722 sits in a hydrophobic pocket formed by Phe330, Leu 342, Leu 689 and Val681 in the β-sheet region of the adjacent monomer. Lys118 and Lys123 for salt bridges with Asp725 and Glu724, respectively. While the structural differences between TRPV2$_{APO\_1}$ and TRPV2$_{CBD\_2}$ are more noticeable than between TRPV2$_{APO\_1}$ and TRPV2$_{CBD\_1}$, nevertheless TRPV2$_{CBD\_2}$ is still in a non-conducting CBD-bound state.

## Opening of the apo TRPV2 selectivity filter

In contrast to TRPV2$_{APO\_1}$, TRPV2$_{CBD\_1}$, and TRPV2$_{CBD\_2}$, TRPV2$_{APO\_2}$ adopted a very different conformation in the lipid environment (*Figure 1* and *Figure 4*). The R.M.S.D. is 1.8 Å when the two apo TRPV2 structures are aligned based on the tetrameric pore of the channel (residues Asp536-His651, covering S5-PH-S6). The most notable difference between these two apo states is the opening of the selectivity filter (*Figure 1C–D*). At the TRPV2$_{APO\_2}$ selectivity filter, Met607 has rotated away from the pore (*Figure 4A*, *Figure 4—figure supplement 1*) and the carbonyl of Gly606 has shifted away from the pore by 5 Å (*Figure 4B*). Half of this distance can be attributed to the outward shift of the whole pore helix, the other half to an additional shift of the loop between the pore helix and S6 (*Figure 4B*). While both Met607 and Gly606 shift away from the pore, Glu609 rotates towards the pore, potentially providing a point of interaction with ions passing through the pore (*Figure 4A–B*).

In TRPV2$_{APO\_2}$, the S5-PH-S6 region of each monomer rotates counterclockwise by 8° independently when viewed from the top (*Figure 4C*) and also moves subtly away from the central axis of the pore, with the pore helix showing an overall movement of 2.3 Å (*Figure 4B*). Despite the rotation of the S5-PH-S6 region, the His651 residues at the bottom of the pore keep the helices together and the interplay between the rotation and the anchoring at His651 causes the base of the S6 helix to curve slightly (*Figure 4D*, *Figure 3—figure supplement 1*). Although this placement of His651 keeps the pore in a non-conducting state, it is easy to imagine that the movement of His651 out of the pore could result in a completely open channel.

The rotations of the pore region of TRPV2$_{APO\_2}$ are transmitted to the ARDs through rotation of the S4-S5 linker and the TRP helix. The S4-S5 linker rotates by 12° to accommodate the rotation of the S5-PH-S6 region (*Figure 4D*). The rotation of the S4-S5 linker is coupled to the TRP helix and

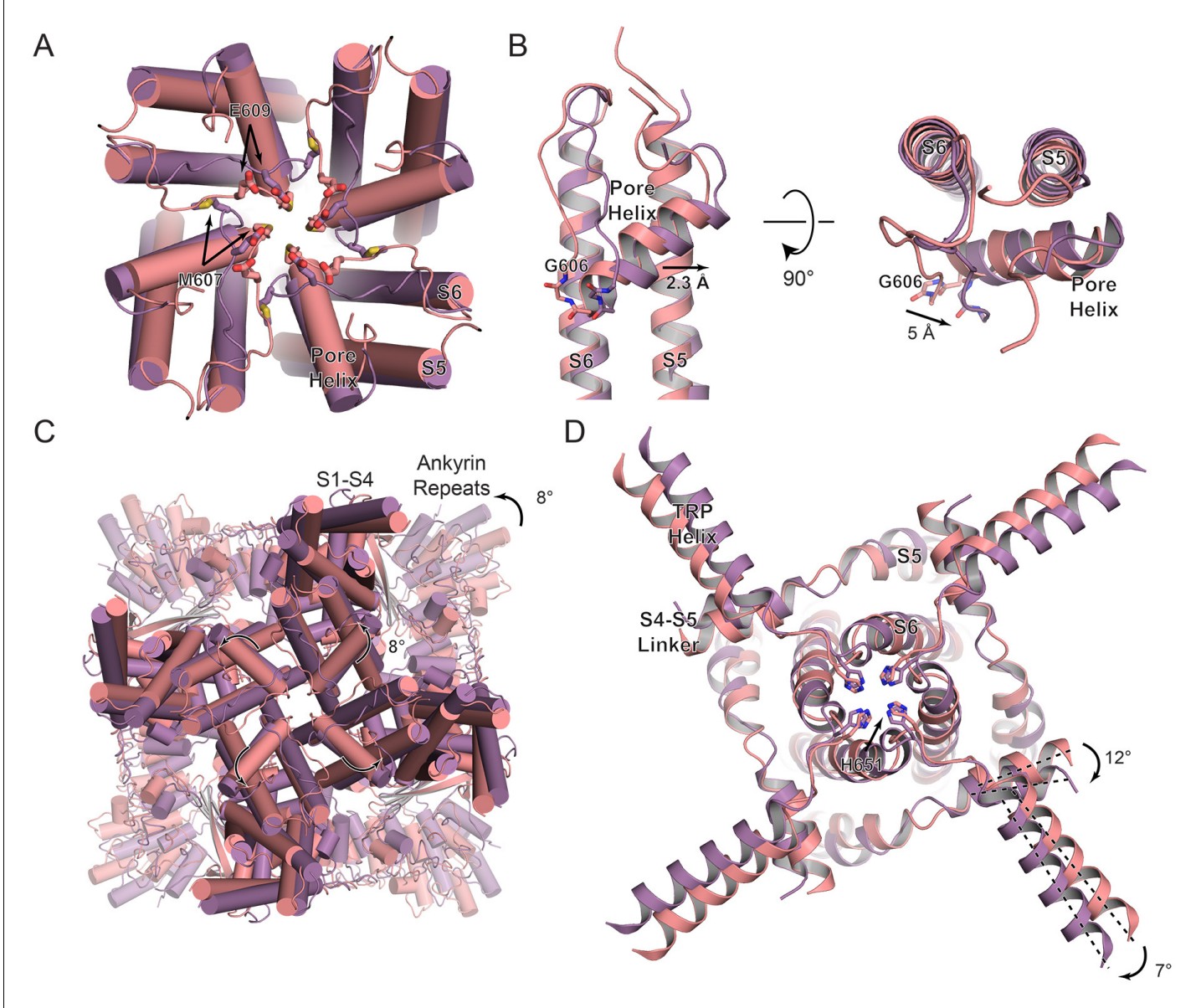

**Figure 4.** Conformational changes between apo TRPV2 states. (A) Overlay of the TRPV2$_{APO\_1}$ (salmon) and TRPV2$_{APO\_2}$ (purple) selectivity filter, viewed from the extracellular side of the membrane. (B) Zoom view of the overlay at S5-PH-S6. (C) Overlay of the TRPV2$_{APO\_1}$ (salmon) and TRPV2$_{APO\_2}$ (purple) structures, viewed from the extracellular side of the membrane. (D) Zoom view from the intracellular side of the membrane of the overlay of the TRPV2$_{APO\_1}$ (salmon) and TRPV2$_{APO\_2}$ (purple) structures. Residues of interest are represented as sticks and labeled. Dashed black lines are references to indicate rotation. The measurements for rotation and lateral shifts are labeled and indicated with black arrows.

DOI: https://doi.org/10.7554/eLife.48792.023

The following figure supplements are available for figure 4:

**Figure supplement 1.** Cryo-EM map density around the selectivity filter of TRPV2$_{APO\_1}$ and TRPV2$_{APO\_2}$.

DOI: https://doi.org/10.7554/eLife.48792.024

**Figure supplement 2.** Conformational changes at the selectivity filter between TRPV1 apo and DkTx/RTX-bound structures in nanodiscs.

DOI: https://doi.org/10.7554/eLife.48792.025

ARD rotation of 7° and 8°, respectively (*Figure 4C–D*, *Figure 3—figure supplement 1*). The flexibility of the loop between the S4-S5 linker and S5 likely contributes to the transformation of individual rotation at the pore monomers to an overall rotation of the whole cytoplasmic face of the channel.

Our newly obtained cryo-EM results are consistent with the previous observations that the purified full-length rat TRPV2 channel displays activity in the presence of ligand as well as exhibiting long spontaneous opening events in the absence of ligands when reconstituted into artificial liposomes (*Huynh et al., 2016*; *Huynh et al., 2014*). Based on these electrophysiological studies and the conformational changes resolved in TRPV2$_{APO\_2}$, we could speculate that the membrane environment permits channel transitions between closed and open states without the addition of specific activators.

## Discussion

Here, we have presented apo and CBD-bound full-length rat TRPV2 structures in a near native lipid environment. Nanodisc technology allowed us to resolve four different full-length rat TRPV2 structures in lipid membranes and observe membrane lipids that associated with the channel. We were able to identify the novel CBD binding site in the TRPV2 channel, which does not correspond to any lipid binding sites in TRPV channels, and determine conformational changes in TRPV2 upon CBD binding. TRPV channels share ~40% sequence homology (*Madej and Ziegler, 2018*) and thermoTRPV channels (TRPV1-TRPV4) have all been shown to have varying levels of modulation by CBD (*Qin et al., 2008*; *De Petrocellis et al., 2012*; *Smart et al., 2000*). TRPV2 interacts with CBD through a hydrophobic pocket located between S5 and S6 helices of adjacent subunits, which is highly conserved among TRPV channels (*Figure 2—figure supplement 3*). This binding site coincides with the location of the π-helix on S6 in all truncated rat TRPV1 structures in nanodiscs (*Gao et al., 2016*), while all the TRPV2 structures reported here had standard α-helixes in this portion of S6. Previous structures of rabbit TRPV2 (*Zubcevic et al., 2019a*; *Zubcevic et al., 2016*) and frog TRPV4 (*Deng et al., 2018*) also had standard α-helical S6 helices. In contrast, TRPV3 studies proposed that a transition between an α-helix and a π-helix on S6 is critical for channel opening (*Singh et al., 2018*; *Zubcevic et al., 2018a*; *Zubcevic et al., 2019b*). Based on our current results, it is unclear whether the presence of the π-helix in TRPV1 and TRPV3 channels would hinder or aid CBD binding to the channel. Further systematic structural analysis of the full-length TRPV1-TRPV4 channels in the presence of CBD would be able to clarify this observation.

Studies on the truncated (*Zubcevic et al., 2018b*) and full-length engineered RTx-sensitive rabbit TRPV2 channel (*Zubcevic et al., 2019a*) suggested that upon RTx binding TRPV2 undergoes a transition between two-fold and four-fold symmetry and that this transition is seen for structures in both amphipols and nanodiscs (*Zubcevic et al., 2019a*). However, we did not see any evidence of two-fold symmetry transition in either of our full-length rat TRPV2 in nanodiscs data sets at any stage of data processing. Additionally, we did not see any indication in our structures that the pore turrets are arranging parallel to the membrane and interact with the lipids in the membrane (*Dosey et al., 2019*). Instead, in all of our structures we saw partial density for the pore turrets extending perpendicularly away from the membrane, confirming similar observations in the full-length engineered RTx-sensitive rabbit TRPV2 channel (*Zubcevic et al., 2019a*).

As all of these full-length rat TRPV2 structures are in static states, the forces and timing of the structural changes we observed are unclear. However, it seems logical to suggest that these changes would originate at the S4-S5 linker, which is the closest divergent region to the site of CBD binding. This region has previously been suggested to function as the gearbox in TRP channels generally (*Hofmann et al., 2017*), mediating the transmission of forces between the pore and the rest of the channel, which is consistent with our observations. This is also consistent with the recent work on the engineered RTx-sensitive rabbit TRPV2 channel (*Zubcevic et al., 2019a*), which showed similar transitions in the S4-S5 linker upon RTx binding (*Zubcevic et al., 2019a*).

Previous work from our group (*Huynh et al., 2016*) and others (*Penna et al., 2006*) has suggested that TRPV2 has some level of constitutive activity. This is supported by the second structure obtained from the apo TRPV2 dataset, TRPV2$_{APO\_2}$. Although this structure features an open selectivity filter, the TRPV2$_{APO\_2}$ pore is still occluded by His651 residues at the bottom of the pore (*Figure 1C–D*). This state may be a snapshot of a semi-open state of the full-length rat TRPV2 channel. Despite the prominent role His651 seems to be playing in rat TRPV2 structures, it should be noted that His651 is not conserved, and in fact the human TRPV2 channel has a serine rather than a histidine in this position (*Figure 2—figure supplement 3*). Similar selectivity filter conformation has been observed in the truncated rat TRPV1 (*Cao et al., 2013*; *Gao et al., 2016*) channel upon RTx

and double-knot toxin (DkTx) application (*Figure 4—figure supplement 2*) and it has been proposed that the selectivity filter plays a critical role in TRPV1 channel gating. All recent TRPV2 cryo-EM studies have made very similar conclusions, suggesting that TRPV1 and TRPV2 share similar gating mechanisms (*Zubcevic et al., 2019a*; *Zubcevic et al., 2016*). The data presented here support this idea and we conclude that the selectivity filter in TRPV2 is important for channel gating.

Recently, it has also been observed that the distal C-terminus in TRPV1-TRPV3 cryo-EM structures can adopt helical or coil conformations (*Zubcevic et al., 2019b*), which may be associated with TRPV channel gating. The distal C-terminus in the truncated apo rat TRPV1 structure and mutant human TRPV3 structures are suggested to be helical, while in multiple wild-type apo TRPV3 structures and in the truncated rabbit TRPV2 channel X-ray structure in the presence of 2 mM $Ca^{2+}$ it is in the coil conformation (*Zubcevic et al., 2019b*). Three of our structures (TRPV2$_{APO\_1}$, TRPV2$_{APO\_2}$, and TRPV2$_{CBD\_1}$) have a helical distal C-terminus, but in TRPV2$_{CBD\_2}$ the distal C-terminus adopts a coil conformation. Additionally, in three of our structures (TRPV2$_{APO\_1}$, TRPV2$_{APO\_2}$, and TRPV2$_{CBD\_1}$) we observe a density that we assigned to the distal N-terminus, that is similar to the mutant human TRPV3 channel (*Zubcevic et al., 2019b*). Nevertheless, in another TRPV3 structure this region was assigned to be an extension of the distal C-terminus (*Singh et al., 2018*). Based on all these recent results (*Singh et al., 2018*; *Zubcevic et al., 2019b*), it is evident that the distal N- and C termini undergo conformation changes in TRPV1-TRPV3 channels, nevertheless further structural analysis of these channels would need to be performed to clarify the role of distal intracellular domains in channels gating.

Overall, our structural studies provided new molecular insights into TRPV2 channel gating and revealed a novel drug binding site in TRPV channels. This new information could be used to guide therapeutic design to treat glioblastoma multiforme and other TRPV2 channel associated pathophysiological process.

## Materials and methods

### Protein expression and purification

The nanodisc reconstituted full-length rat TRPV2 was expressed and purified as previously published (*Huynh et al., 2016*), with minor modifications. The membranes expressing rat TRPV2 were solubilized in 20 mM HEPES pH 8.0, 150 mM NaCl, 5% glycerol, 0.087% LMNG, 2 mM TCEP, and 1 mM PMSF for 1 hr. Insoluble material was removed via ultra-centrifugation at 100,000 x g and the solubilized TRPV2 was purified by binding to 1D4 antibody coupled CnBr-activated Sepharose beads. The beads were washed with Wash Buffer containing 0.006% DMNG and the protein was eluted with Wash Buffer containing 0.006% DMNG and 3 mg/ml 1D4 peptide. The protein was then reconstituted into nanodiscs in a 1:1:200 ratio of TRPV2:MSP2N2:soy polar lipids (Avanti). MSP2N2 was expressed in BL21 (DE3) cells and purified via affinity chromatography as previously described (*Hughes et al., 2018b*). Lipids were dried under a nitrogen flow prior to reconstitution and resuspended in Wash Buffer containing DMNG in a 1:2.5 ratio (soy polar lipids:DMNG). The nanodisc reconstitution mixture was incubated at 4°C for 30 mins. Bio-Beads (Bio-Beads SM-2 Absorbent Media, Bio-Rad) were added to the reconstitution mixture for 1 hr then the reaction mixture was transferred to fresh Bio-Beads and the mixture was incubated overnight. Nanodisc reconstituted TRPV2 was further purified using size-exclusion chromatography (Superose 6, GE Healthcare) in Wash Buffer. Protein eluted from the column was concentrated to 2 mg/mL for use in vitrification.

### Cryo-EM data collection

For apo TRPV2, fluorinated Fos-choline eight was added to the sample to a final concentration of 3 mM just before blotting. This sample was double blotted (3.5 µl per blot) onto glow discharged 200 mesh Quantifoil 1.2/1.3 grids (Quantifoil Micro Tools) at 4°C and 100% humidity and plunge frozen in liquid ethane cooled to the temperature of liquid nitrogen (Thermo Fisher Vitrobot). Cryo-EM images were collected using a 300kV Thermo Fisher Krios microscope equipped with a Gatan K3 direct detector camera in super resolution mode. 40 frame movies were collected with a total dose of 50 e/$Å^2$ and a super resolution pixel size of 0.53 Å/pix. Defocus values of the images range from −0.8 to −3.0 µm.

For CBD-bound TRPV2, prior to preparing cryo-EM grids purified TRPV2 was incubated with 30 µM CBD for 30 mins. Fluorinated Fos-choline eight was added to the sample to a final concentration of 3 mM just before blotting. This sample was blotted (3 µl per blot) onto glow discharged 200 mesh Quantifoil 1.2/1.3 grids (Quantifoil Micro Tools) at 4°C and 100% humidity and plunge frozen in liquid ethane cooled to the temperature of liquid nitrogen (Thermo Fisher Vitrobot). Cryo-EM images were collected using a 300kV Thermo Fisher Krios microscope equipped with a Gatan K3 direct detector camera in counting mode. 40 frame movies were collected with a total dose of 50 e/$Å^2$ and a pixel size of 1.06 Å/pix. Defocus values of the images range from −0.8 to −3.0 µm.

## Image processing

For both datasets, all data processing was conducted in RELION 3.0 (*Scheres, 2012*; *Scheres, 2016*; *Zivanov et al., 2018*). The movie frames were aligned using MotionCor2 (*Zheng et al., 2017*), with the apo dataset being binned by two to a pixel size of 1.06 Å/pix, to compensate for beam-induced motion. Defocus values of the motion corrected micrographs were estimated using Gctf (*Zhang, 2016*). At this point, suboptimal micrographs were removed from each dataset based on manual inspection and statistics generated by Gctf. Initially, a subset of 100 micrographs were picked from each with Laplacian-of-Gain auto-picking, which yielded on the order of 10 s of thousands of particles for each dataset. Each set of particles were subsequently sorted into 2D classes to generate templates for standard auto-picking for each dataset. For the apo dataset, auto-picking of 10,548 micrographs resulted in ~1.2 million auto-picked particles, for the CBD-bound dataset, auto-picking of 5024 micrographs resulted in ~2.3 million auto-picked particles. These were then subjected to 2D classification to remove suboptimal particles and false positive hits. The best 596,859 particles from the apo and 710,728 from the CBD-bound dataset were refined without applied symmetry. The initial model for these refinements was created from the density map of the previously published full length TRPV2 (*Huynh et al., 2016*) (EMB-6580) filtered to 60 Å. These initially refined particles were then subjected to 3D classification into eight classes without angular sampling and no applied symmetry, using a soft mask made from the initial 3D refinement. For both datasets, this classification yielded a single, clear best class with good TRP channel features. The best classes from the apo dataset had 117, 582 particles and from the CBD-bound dataset had 153,464 particles. Each set of particles were refined with no applied symmetry and tested for symmetry using the Map Symmetry tool in PHENIX (*Adams et al., 2002*), which assigned C4 symmetry to both maps. These particles were then refined again with C4 symmetry before being subjected to CTF refinement, Bayesian polishing, and an additional round of 2D classification to further remove noise. After this treatment, the apo dataset had 96,161 polished particles and the CBD-bound dataset had 125,038 polished particles. Each set of particles was then refined with C4 symmetry followed by 3D classification without angular sampling into five classes for the apo dataset and six classes for the CBD-bound dataset, using a mask of the full TRP channel but excluding density for the nanodisc. Sorting for both datasets yielded three best classes, and in both datasets two of those classes were sufficiently similar to combine. This resulted in two final sets of particles for each dataset, 42,407 (TRPV2$_{APO\_1}$) and 14,332 (TRPV2$_{APO\_2}$) for the apo dataset and 23,944 (TRPV2$_{CBD\_1}$) and 36,153 (TRPV2$_{CBD\_2}$) for the CBD-bound dataset. Each of these sets of particles was again subjected to CTF refinement and Bayesian polishing before a final refinement with C4 symmetry, followed by post-processing using a mask of the channel that excluded the nanodisc cloud yielding maps at 3.7 Å (TRPV2$_{APO\_1}$), 4.0 Å (TRPV2$_{APO\_2}$), 3.4 Å (TRPV2$_{CBD\_1}$) and 3.2 Å (TRPV2$_{CBD\_2}$). Local resolution maps were generated using the RELION (*Zivanov et al., 2018*; *Scheres, 2016*; *Scheres, 2012*) implementation of local resolution estimation.

## Model building

The previously determined full-length rat TRPV2 structure (PDB: 5HI9) was employed as the initial starting model and docked into the TRPV2$_{APO\_1}$ map. This model was then manually adjusted in COOT (*Emsley and Cowtan, 2004*) and refined using phenix.real_space_refine from the PHENIX software package (*Adams et al., 2002*) with four-fold NCS constraints. The model was subjected to iterative rounds of manual model fitting followed by real-space refinement and sidechains with insufficient density were removed. The disconnected 16 residue loop attributed to Gln30-Asn45 between the ARDs and the β-sheet region in the TRPV2$_{APO\_1}$ model was initially built as alanines, with residue

identities determined based on distinctive density for Met35 and Phe39 along with kinking of the backbone that could be traced to a distinctive pattern of three proline residues at the N-terminus of the protein. Models fit into the other three maps started from the final $_{APO\_1}$ model, followed by the same process of manual adjustment and refinement. The ligand restraint file for CBD was generated using the eLBOW tool from the PHENIX software package (*Moriarty et al., 2009*).

Each final model was randomized by 0.5 Å in PHENIX (*Adams et al., 2002*) and refined against a single half map. These models were converted into volumes in Chimera (*Pettersen et al., 2004*) and then EMAN2 (*Ludtke et al., 1999*) was used to generate FSC curves between these models and each half map as well as between each final model and the final maps. HOLE was used to generate the pore radii (*Smart et al., 1996*). Pymol and Chimera (*Pettersen et al., 2004*) were used to align models and maps and to make figures.

## HEK293 cell culture, mutagenesis and transfection

Human Embryonic Kidney 293T (HEK293T) cells were purchased from American Type Culture Collection (ATCC), Manassas, VA, (catalogue # CRL-3216), RRID:CVCL_0063 and tested regularly for mycoplasma contamination. Passage number of the cells was monitored, and cells were used up to passage number 25–30. The cells were maintained in Dulbecco's Modified Eagle's Medium (DMEM) (ATCC, catalogue # 30–2002) supplemented with 10% (v/v) fetal bovine serum (FBS), GlutaMAX-I (Gibco, catalogue # 35050), 100 IU/ml penicillin and 100 μg/ml streptomycin and were kept in a tissue-culture incubator with 5% $CO_2$ at 37°C. The cells were transiently transfected with cDNA encoding the rat TRPV2 (rTRPV2-WT, Leu541F-L631Phe or Val635Phe mutant), in the pcDNA3 vector and pEYFP in ratio 1:0.1 using the Effectene reagent (Qiagen) according manufacturer's protocol and used in experiments 48–72 hr later. Point mutations were introduced using the QuickChange Mutagenesis Kit (Agilent).

## TRPV2 channel electrophysiology

Whole-cell patch clamp measurements were performed as described earlier (*Badheka et al., 2015*). Measurements were carried out on YFP positive cells, in an extracellular solution containing (in mM) 137 NaCl, 5 KCl, 1 $MgCl_2$, 10 HEPES and 10 glucose, pH 7.4. The intracellular solution contained (in mM) 135 Cs-Metanesulfonate, 1 $MgCl_2$, 10 HEPES, 5 EGTA, 4 NaATP (pH 7.25). Patch clamp pipettes were prepared from borosilicate glass capillaries (Sutter Instruments) using a P-97 pipette puller (Sutter Instrument) and had a resistance of 4–6 MΩ. In all experiments after formation of gigaohm-resistance seals, the whole-cell configuration was established and currents were recorded using a ramp protocol from −100 mV to +100 mV over 500 ms preceded by a −100 mV step for 200 ms; the holding potential was −60 mV, and this protocol was applied once every 2 s. The currents were measured with an Axopatch 200B amplifier, filtered at 5 kHz, and digitized through the Digidata 1440A interface. In all experiments, cells that had a passive leak current more than 100 pA were discarded. Data were collected and analyzed with the PClamp10.6 (Clampex) acquisition software (Molecular Devices, Sunnyvale, CA), and further analyzed and plotted with Origin 8.0 (Microcal Software Inc, Northampton, MA, USA).

## Data availability

The cryo-EM density maps and the atomic coordinates of the apo and both CBD-bound full-length TRPV2 channels in nanodiscs are deposited into the Electron Microscopy Data Bank and Protein Data Bank under accession codes EMD-20677 and PDB 6U84 (TRPV2$_{APO\_1}$), EMD-20678 and PDB 6U86 (TRPV2$_{APO\_2}$), EMD-20686 and PDB 6U8A (TRPV2$_{CBD\_1}$), and EMD-20682 and PDB 6U88 (TRPV2$_{CBD\_2}$).

## Acknowledgements

We thank David Lodowski at Case Western Reserve University for help in the early stage of the project. We thank Sudha Chakrapani at Case Western Reserve University for assistance with MSP2N2 expression and purification. We thank Sabine Baxter for assistance with hybridoma and cell culture at the University of Pennsylvania Perelman School of Medicine Cell Center Services Facility. We acknowledge the use of instruments at the Electron Microscopy Resource Lab and at the Beckman Center for Cryo Electron Microscopy at the University of Pennsylvania Perelman School of Medicine.

We also thank Darrah Johnson-McDaniel for assistance with Krios microscope operation. This work was supported by grants from the National Institute of Health (R01GM103899 and R01GM129357 to VYM-B, R01NS055159 and R01GM093290 to TR).

## Additional information

### Competing interests

Yuhang Liu, Seungil Han: is affiliated with Pfizer Research and Development. The author has no financial interests to declare. The other authors declare that no competing interests exist.

### Funding

| Funder | Grant reference number | Author |
| --- | --- | --- |
| National Institutes of Health | R01GM129357 | Vera Y Moiseenkova-Bell |
| National Institutes of Health | R01GM103899 | Vera Y Moiseenkova-Bell |
| National Institutes of Health | R01 NS055159 | Tibor Rohacs |
| National Institutes of Health | R01GM093290 | Tibor Rohacs |

The funders had no role in study design, data collection and interpretation, or the decision to submit the work for publication.

### Author contributions

Ruth A Pumroy, Data curation, Formal analysis, Validation, Investigation, Visualization, Methodology, Writing—original draft, Writing—review and editing; Amrita Samanta, Yevgen Yudin, Investigation; Yuhang Liu, Resources, Investigation; Taylor ET Hughes, Investigation, Writing—review and editing; Siyuan Zhao, Formal analysis, Validation, Investigation, Visualization; Tibor Rohacs, Resources, Supervision, Funding acquisition, Writing—review and editing; Seungil Han, Resources, Supervision; Vera Y Moiseenkova-Bell, Conceptualization, Supervision, Funding acquisition, Methodology, Writing—original draft, Project administration, Writing—review and editing

### Author ORCIDs

Ruth A Pumroy https://orcid.org/0000-0002-6200-6083
Yuhang Liu http://orcid.org/0000-0001-6844-7480
Tibor Rohacs http://orcid.org/0000-0003-3580-2575
Seungil Han https://orcid.org/0000-0002-1070-3880
Vera Y Moiseenkova-Bell https://orcid.org/0000-0002-0589-4053

### Decision letter and Author response

Decision letter https://doi.org/10.7554/eLife.48792.044
Author response https://doi.org/10.7554/eLife.48792.045

## Additional files

### Supplementary files

• Transparent reporting form
DOI: https://doi.org/10.7554/eLife.48792.026

### Data availability

cryoEM maps have been deposited in the Electron Microscopy Data Bank under the following accession codes: EMD-20677, EMD-20678, EMD-20686 and EMD-20682. The models built into the cryoEM maps have been deposited into the Protein Data Bank under the following accession codes: 6U84, 6U86, 6U8A and 6U88. The maps and models analyzed in this study are included with the manuscript and supporting files.

The following datasets were generated:

| Author(s) | Year | Dataset title | Dataset URL | Database and Identifier |
|---|---|---|---|---|
| Pumroy RA, Moiseenkova-Bell VY | 2019 | Apo full-length rat TRPV2 in nanodiscs, state 1 | https://www.ebi.ac.uk/pdbe/entry/emdb/EMD-20677 | Electron Microscopy Data Bank, EMD-20 677 |
| Pumroy RA, Moiseenkova-Bell VY | 2019 | Apo full-length rat TRPV2 in nanodiscs, state 1 | https://www.rcsb.org/structure/6U84 | Protein Data Bank, PDB 6U84 |
| Pumroy RA, Moiseenkova-Bell VY | 2019 | Apo full-length rat TRPV2 in nanodiscs, state 2 | https://www.ebi.ac.uk/pdbe/entry/emdb/EMD-20678 | Electron Microscopy Data Bank, EMD-20 678 |
| Pumroy RA, Moiseenkova-Bell VY | 2019 | Apo full-length rat TRPV2 in nanodiscs, state 2 | https://www.rcsb.org/structure/6U86 | Protein Data Bank, PDB 6U86 |
| Pumroy RA, Moiseenkova-Bell VY | 2019 | CBD-bound full-length rat TRPV2 in nanodiscs, state 1 | https://www.ebi.ac.uk/pdbe/entry/emdb/EMD-20686 | Electron Microscopy Data Bank, EMD-20 686 |
| Pumroy RA, Moiseenkova-Bell VY | 2019 | CBD-bound full-length rat TRPV2 in nanodiscs, state 1 | https://www.rcsb.org/structure/6U8A | Protein Data Bank, PDB 6U8A |
| Pumroy RA, Moiseenkova-Bell VY | 2019 | CBD-bound full-length rat TRPV2 in nanodiscs, state 2 | https://www.ebi.ac.uk/pdbe/entry/emdb/EMD-20682 | Electron Microscopy Data Bank, EMD-20 682 |
| Pumroy RA, Moiseenkova-Bell VY | 2019 | CBD-bound full-length rat TRPV2 in nanodiscs, state 2 | https://www.rcsb.org/structure/6U88 | Protein Data Bank, PDB 6U88 |

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
