## [Decision Letter]

Thank you for submitting your article "Molecular mechanism of TRPV2 channel modulation by cannabidiol" for consideration by *eLife*. Your article has been reviewed by three peer reviewers, including Leon D Islas as the Reviewing Editor, and the evaluation has been overseen by a Reviewing Editor and Olga Boudker as the Senior Editor.

Given the list of essential revisions, including new experiments, the editors and reviewers invite you to respond with an action plan and timetable for the completion of the additional work. We plan to share your responses with the reviewers and then issue a binding recommendation.

Summary:

This manuscript by Pumroy et al., describes two new structures of the TRPV2 ion channel, both in detergent and in nanodiscs. It also provides evidence of a new binding site for canabidiol (CBD), a pharmacological activator of this channel. This is an important finding, since it can provide a foundation to understand the actions of CBD on TRPV2 and other TRPV channels and the potential therapeutic actions of this drug, as well as the molecular mechanisms of activation of TRPV2.

Essential revisions:

Although the reviewers agree that the findings presented in your manuscript are of potential relevance to the field, there are a number of problems that need to be addressed for the manuscript to be acceptable in *eLife*.

1) The three reviewers agree that the conclusions are not strongly supported by the data. The resolution of the refines structures is poor and thus, additional experimental information is needed to support the claims of a new binding site for CBD. Phrases like "The CBD-bound TRPV2 structure in nanodiscs clearly revealed that CBD interacts…" (Introduction) must be avoided through the manuscript.

2) Most of the conformational changes observed between the structures are small, and in fact smaller than the resolution of the structures. In order to provide a strong support for these proposed structural changes, the authors should include the density maps in their structural comparisons. These include the changes in the S1-S4 domains in Figure 1D; the changes in S4-S5 linker and TRP helix in Figure 2B and C; the reorientation of Y525 in Figure 2D,E; the presence of a π-helix in the S4-S5 linker shown in Figure 4, etc. I would not think that the presence of a π-helix can be formally argued at a resolution close to 4 Å. Again the authors should extensively curb their language when describing the small changes that are observed, unless they are unambiguously supported by the data and the experimental densities, and avoid phrases like "CBD induced dramatic conformational changes to S1-S4, the TRP helix, and ARDs…" (subsection “Detergent-solubilized TRPV2 in CBD-bound state”).

3) The static cryo-EM maps do not provide any information about the causality or timing of events, or whether some regions exert force to push or pull on other regions of the protein. These are all entirely hypothetical interpretations, and should be described as such, ideally together with additional experimental data in support of the proposed mechanism. The authors should extensively edit the text to reflect this.

4) The experimental evidence presented to argue that specific residues at the "entrance of the binding site" have been located is at best marginal. The residues probed are far removed from the proposed binding site and thus the results do not demonstrate that the binding site is located where the authors suggest. Finally, only using calcium imaging is a very poor experimental choice. The authors have identified a large number of residues proposed to interact with CBD at its binding site. Given that this is the central finding of the manuscript, as most conformational changes in the protein observed in the structures are very small, it is essential that the authors provide additional mutagenesis results to support it. This would ideally involve high-quality electrophysiological experiments for a systematic evaluation of the effects of various residue substitutions within the pocket, showing that the effect of CBD can be selectively affected relative to 2-APB, and that the effects of the mutations are as predicted from the properties of the binding site.

5) For the apo TRPV2 structure, the authors state "Postprocessing yielded a 4.3Å resolution map as estimated by Rmeasure." This method of estimating resolution has not been recommended since 2012 and has been superseded by the "gold-standard" FSC 0.143 protocol as detailed by Henderson (2012) and Scheres (2012). Although it is not explicitly stated for the other structures, it is assumed this method for estimating resolution was also used. The authors should report the resolution of their structures using gold-standard FSC criterion, as recommended by the EM Validation Task Force.

6) In accordance with recommended guidelines for publishing EM structures as outlined by the EM Validation Task Force, the authors are encouraged to deposit their unfiltered half maps, unsharpened final map, and mask(s) used for refinement/resolution estimation to the Electron Microscopy Data Bank (EMDB).

7) Each of the three maps discussed in this manuscript arise from different magnifications. Small errors in the pixel size calibrations for each of these magnifications can lead to an apparent "change in magnification" of the final 3D volume which can introduce small, consistent errors that can make comparisons across magnifications tenuous, especially when the extent of conformational changes, like those discussed in this manuscript, are small (i.e., <2 Å). Additionally, errors in pixel size can also lead to elevated MolProbity clashscores, like those observed for the apo TRPV2 structure, or require non-optimal refinement weights leading to unusually low (or high) bond length RMSDs. The authors are encouraged to verify the pixel size for each magnification, as implemented in Rosetta or other programs.

8) None of the figures in this manuscript detail interactions with the hydroxyl groups of CBD but rather focus solely on the hydrophobic interactions that could stabilize the proposed binding pose of CBD. Importantly, in the CBD-nanodisc structure, the proposed CBD binding pose places the hydroxyl groups directed at the backbone atoms of Leu631 or the phenyl ring of Phe540, neither of which form favorable hydrogen bonding interactions. This raises the question of how stable was the binding pose during refinement? What are the observed clashes/non-favorable interactions within 5 Å of the modeled CBD? Additionally, it appears CBD binds in a mixed pose with different occupancies. How does CBD refine when flipped 180° about the benzyl ring plane?

9) In accordance with recommended guidelines for publishing EM structures as outlined by the EM Validation Task Force, a detailed processing workflow for each structure should be provided in the supplemental figures.

10) Recent publications detailing the activation mechanisms of TRPV family members have observed deviations from the canonical C4 symmetric state of TRPV family members to C2 symmetric states upon agonist binding or desensitization (Zubcevic et al., 2018 and 2019). Importantly, this was also observed for TRPV2 (Zubcevic et al., 2019) from a different organism (rabbit). The authors are encouraged to provide a detailed analysis of the symmetry observed in their structures to ensure the TRPV2 molecules in their data are, in fact, C4-symmetric. This is especially important given how few particles are maintained in the final reconstructions after 2D classification.

11) The authors describe the CBD-nanodisc structure as a potential intermediate, yet, this is the only CBD-bound structure where the ligand can be "fully" observed. This is particularly confusing given the amount of excess CBD added during complex preparation exceeded the concentration of CBD added to the CBD-detergent sample. Therefore, one would expect the CBD-detergent structure to represent a potential intermediate with low CBD occupancy. Please clarify.

12) The authors claim that their CBD-bound TRPV2 structure (in detergent) adopts a "partially open" conformation, yet, M645 and V649 lie along the pore in positions that render the lower gate impermeable, even to fully dehydrated cations. If anything, addition of CBD results in a rearrangement of V649, placing it closer to the central pore axis and further blocking ion permeation. Despite these observations, it is unclear why the authors speculate that this structure was in the "partially open" conformation.

13) The authors state that CBD-binding leads to opening of the selectivity filter, yet, the largest diameter of the filter is only ~2 Å in diameter, which is insufficient to even allow passage of dehydrated Ca^2+^ ions (diameter = 2.28 Å). Furthermore, it has been speculated for other TRPV family members that ions pass through the selectivity filter in a partially dehydrated form, which would require a pore radius of 1.5-3.5 Å.

14) The authors make a comparison between the Y575-M672 stabilizing interactions in their CBD-nanodisc structure to the recently-published TRPV3 structures. As the observation of the π-helix in S4-S5 is different between the two structures, i.e., present in the apo TRPV3 structure but missing in the apo TRPV2 structure, this "stabilizing" interaction is not 100% conserved. I encourage the authors to further develop this section of the manuscript to better clarify their message.

[Editors' note: further revisions were requested prior to acceptance, as described below.]

Thank you for re-submitting your article "Molecular mechanism of TRPV2 channel modulation by cannabidiol" for consideration by *eLife*. Your revised article has been favourably valuated by a Reviewing Editor and Olga Boudker as the Senior Editor.

The manuscript has been improved, but there are some remaining issues that need to be addressed before acceptance.

Summary:

The manuscript by the group of Moiseenkova-Bell presents the identification of a CBD binding site in TRPV2 channels. The location of this site is novel, as it is located between the S5 and S6 of different subunits. CBD is an important compound of high clinical interest and this manuscript makes an important contribution to our understanding of the modulation of TRPV channels by compounds of cannabinoid origin, in particular CBD. The new data is of high quality and the identification of CBD bound to TRPV2 is solidly presented.

Essential revisions:

The reviewers feel that the current version of the manuscript is much improved and the authors have made a good effort to address the concerns expressed in the previous revision. However, the reviewers also think that the electrophysiological data is still lacking and does not strongly support the identification of the binding site. In particular, having a similar current magnitude in response to 2-APB between WT and mutant does not constitute evidence that the mutations do not affect sensitivity to 2-APB, because the current magnitude does not distinguish between changes in open probability or level of channel expression. This experiment should therefore be removed from the manuscript.

The effect of the double mutation on the CBD to 2-APB current ratio is negligible, despite it being statistically significant. This data can be included in the manuscript, but the conclusion that this mutation identifies the binding site for CBD should be toned down.

---

## [Author Response]

[Editors' note: the authors’ plan for revisions was approved and the authors made a formal revised submission.]

Essential revisions:Although the reviewers agree that the findings presented in your manuscript are of potential relevance to the field, there are a number of problems that need to be addressed for the manuscript to be acceptable in eLife.1) The three reviewers agree that the conclusions are not strongly supported by the data. The resolution of the refines structures is poor and thus, additional experimental information is needed to support the claims of a new binding site for CBD. Phrases like "The CBD-bound TRPV2 structure in nanodiscs clearly revealed that CBD interacts…" (Introduction) must be avoided through the manuscript.

We were able to obtain full-length rat TRPV2 structures in the absence (apo state) and in the presence of 30μM CBD (CBD-bound state) in lipid nanodiscs. In the revised manuscript we report two TRPV2 structures in the apo state determined to 3.7 Å (state 1) and 4.0 Å (state 2) and two TRPV2 structures in 30μM CBD-bound state determined to 3.4 Å (state 1) and 3.2 Å (state 2). These improved rat full-length TRPV2 cryo-EM structures in nanodiscs provide clearer evidence for the location of the CBD binding site between S5 and S6 helices of adjacent subunits than our previous data. Still, we agree with reviewers and we toned down the language in the manuscript to avoid possible overstatements.

2) Most of the conformational changes observed between the structures are small, and in fact smaller than the resolution of the structures. In order to provide a strong support for these proposed structural changes, the authors should include the density maps in their structural comparisons. These include the changes in the S1-S4 domains in Figure 1D; the changes in S4-S5 linker and TRP helix in Figure 2B and C; the reorientation of Y525 in Figure 2D,E; the presence of a π-helix in the S4-S5 linker shown in Figure 4, etc. I would not think that the presence of a π-helix can be formally argued at a resolution close to 4 Å. Again the authors should extensively curb their language when describing the small changes that are observed, unless they are unambiguously supported by the data and the experimental densities, and avoid phrases like "CBD induced dramatic conformational changes to S1-S4, the TRP helix, and ARDs…" (subsection “Detergent-solubilized TRPV2 in CBD-bound state”).

The manuscript has been carefully revised and all figures that describe structural changes in the rat full-length TRPV2 channel include figure supplements to show density maps for these proposed changes. Again, we agree with reviewers and we toned down the language in the manuscript to avoid possible overstatements.

3) The static cryo-EM maps do not provide any information about the causality or timing of events, or whether some regions exert force to push or pull on other regions of the protein. These are all entirely hypothetical interpretations, and should be described as such, ideally together with additional experimental data in support of the proposed mechanism. The authors should extensively edit the text to reflect this.

Again, we agree with reviewers and we toned down the language in the manuscript to avoid possible overstatements.

4) The experimental evidence presented to argue that specific residues at the "entrance of the binding site" have been located is at best marginal. The residues probed are far removed from the proposed binding site and thus the results do not demonstrate that the binding site is located where the authors suggest. Finally, only using calcium imaging is a very poor experimental choice. The authors have identified a large number of residues proposed to interact with CBD at its binding site. Given that this is the central finding of the manuscript, as most conformational changes in the protein observed in the structures are very small, it is essential that the authors provide additional mutagenesis results to support it. This would ideally involve high-quality electrophysiological experiments for a systematic evaluation of the effects of various residue substitutions within the pocket, showing that the effect of CBD can be selectively affected relative to 2-APB, and that the effects of the mutations are as predicted from the properties of the binding site.

In the revised manuscript, we now describe the rat full-length TRPV2 apo and 30μM CBD-bound structures in nanodiscs. All cryo-EM data were collected using the same Krios microscope. Based on the analysis of the half maps for apo and 30μM CBD-bound structures in nanodiscs, we assigned CBD to the density that is positioned between S5 and S6 helices of adjacent subunits. This density is absent in rat full-length TRPV2 apo structures in nanodiscs.

We originally hypothesized that mutating Leu541, Leu631 and Val635, the residues that are part of the proposed CBD binding pocket (see Figure 3), to bulky hydrophobic residues like phenylalanine might alter the entrance of CBD into the pocket. We now performed whole cell patch clamp recordings on wild type and mutant TRPV2 channels. We now show that the ratio of currents induced by 20 μM CBD and 100 μM 2-APB was significantly higher in cells transfected with the Leu541Phe-Leu631Phe mutant compared to wild type TRPV2, but the 100 μM 2-APB induced currents were of similar amplitude in both channels (see Figure 2—figure supplement 4). This data and our new cryo-EM results suggest that we assigned CBD density appropriately and it interacts with TRPV2 through the pocket located between S5 and S6 helices of adjacent subunits.

Neither 20 μM CBD not 100 μM 2-APB induced any current in non-transfected cells and in cells transfected with the Val635Phe mutant, indicating that the latter mutant was non-functional (data not shown). The CBD binding site is located between S5 and S6 helices of adjacent subunits in TRPV2. Val635 located on the interface of these subunits and based on these results, we decided not to mutate more residues that are located on the interface of S5 and S6 helices of adjacent subunits in TRPV2, as we thought these mutants will likely affect tetramer formation and produce non-functional channels.

5) For the apo TRPV2 structure, the authors state "Postprocessing yielded a 4.3Å resolution map as estimated by Rmeasure." This method of estimating resolution has not been recommended since 2012 and has been superseded by the "gold-standard" FSC 0.143 protocol as detailed by Henderson (2012) and Scheres (2012). Although it is not explicitly stated for the other structures, it is assumed this method for estimating resolution was also used. The authors should report the resolution of their structures using gold-standard FSC criterion, as recommended by the EM Validation Task Force.

All cryo-EM data is now presented in accordance with recommendations from cryo-EM Validation Task Force.

6) In accordance with recommended guidelines for publishing EM structures as outlined by the EM Validation Task Force, the authors are encouraged to deposit their unfiltered half maps, unsharpened final map, and mask(s) used for refinement/resolution estimation to the Electron Microscopy Data Bank (EMDB).

All of the requested data is now deposited into the EMDB.

7) Each of the three maps discussed in this manuscript arise from different magnifications. Small errors in the pixel size calibrations for each of these magnifications can lead to an apparent "change in magnification" of the final 3D volume which can introduce small, consistent errors that can make comparisons across magnifications tenuous, especially when the extent of conformational changes, like those discussed in this manuscript, are small (i.e., <2 Å). Additionally, errors in pixel size can also lead to elevated MolProbity clashscores, like those observed for the apo TRPV2 structure, or require non-optimal refinement weights leading to unusually low (or high) bond length RMSDs. The authors are encouraged to verify the pixel size for each magnification, as implemented in Rosetta or other programs.

In the last two months, we were able to obtain full-length rat TRPV2 structures in the absence (apo states) and in the presence of 30μM CBD (CBD-bound states) in lipid nanodiscs using the same Krios microscope at identical magnifications and the same pixel size. Additionally, the models built for each structure were compared to a crystal structure of the TRPV2 Ankyrin Repeat Domain (PDB 2ETB) to verify the accuracy of pixel size.

8) None of the figures in this manuscript detail interactions with the hydroxyl groups of CBD but rather focus solely on the hydrophobic interactions that could stabilize the proposed binding pose of CBD. Importantly, in the CBD-nanodisc structure, the proposed CBD binding pose places the hydroxyl groups directed at the backbone atoms of Leu631 or the phenyl ring of Phe540, neither of which form favorable hydrogen bonding interactions. This raises the question of how stable was the binding pose during refinement? What are the observed clashes/non-favorable interactions within 5 Å of the modeled CBD? Additionally, it appears CBD binds in a mixed pose with different occupancies. How does CBD refine when flipped 180° about the benzyl ring plane?

CBD was very stable in this position during PHENIX refinement. Examination of the hydroxyl groups using MolProbity and Coot did not show clashes between the CBD hydroxyl groups and TRPV2, and instead suggested hydrogen bonds could be made with the backbone of both S5 and S6, potentially by forming bifurcated hydrogen bonds with the helical backbone. Bifurcation of the hydrogen bonds of an α-helix backbone by interaction with thiol or hydroxyl groups of sidechains like cysteine or serine, particularly in the context of helices in hydrophobic environments, has been established for many years (Gray and Matthews, 1984; Feldblum and Arkin, 2014). It seems feasible that the hydroxyl groups of CBD could be interacting with the backbones of S5 (between the carbonyl of Leu537 and the amide of Leu541) and S6 (between the carbonyl of Leu631 and amide of Val635) in the same way, particularly given that the rest of the pocket is very hydrophobic. We included this explanation in the text.

As far as the orientation of the ligand, the density on the interior of the pocket is quite strong and has two lobes, which makes us think CBD could not be flipped end to end with the acyl tail in the density inside of the pocket. The larger hydrophobic ring end of the molecule could potentially be rotated in the pocket 180° degrees, but it clashed with the Leu537 sidechain in the TRPV2_CBD_1_ structure, indicating the original pose as the optimal fit for this data (see Figure 2—figure supplement 1).

9) In accordance with recommended guidelines for publishing EM structures as outlined by the EM Validation Task Force, a detailed processing workflow for each structure should be provided in the supplemental figures.

All cryo-EM data is now presented in accordance with recommendations from cryo-EM Validation Task Force. Detailed processing workflow is added in the figure supplements.

10) Recent publications detailing the activation mechanisms of TRPV family members have observed deviations from the canonical C4 symmetric state of TRPV family members to C2 symmetric states upon agonist binding or desensitization (Zubcevic et al., 2018 and 2019). Importantly, this was also observed for TRPV2 (Zubcevic et al., 2019) from a different organism (rabbit). The authors are encouraged to provide a detailed analysis of the symmetry observed in their structures to ensure the TRPV2 molecules in their data are, in fact, C4-symmetric. This is especially important given how few particles are maintained in the final reconstructions after 2D classification.

All four non-symmetrized C1 rat full-length TRPV2 maps in nanodiscs presented in this revised manuscript were evaluated by the Map Symmetry function in Phenix and C4 symmetry yielded the highest correlation coefficient for all presented maps.

All published cryo-EM maps for TRPV1-TRPV6 channels included between 2% to 20% of total particles in final reconstructions after 2D classification. Our cryo-EM data falls in the same range of total TRPV2 channel particles in the final maps.

11) The authors describe the CBD-nanodisc structure as a potential intermediate, yet, this is the only CBD-bound structure where the ligand can be "fully" observed. This is particularly confusing given the amount of excess CBD added during complex preparation exceeded the concentration of CBD added to the CBD-detergent sample. Therefore, one would expect the CBD-detergent structure to represent a potential intermediate with low CBD occupancy. Please clarify.

Due to the poor resolution of our detergent structures, we removed them from this version of the manuscript. We now only report data for rat full length TRPV2 in nanodiscs that could be directly compared to each other and text in the manuscript has been revised to reflect these changes.

12) The authors claim that their CBD-bound TRPV2 structure (in detergent) adopts a "partially open" conformation, yet, M645 and V649 lie along the pore in positions that render the lower gate impermeable, even to fully dehydrated cations. If anything, addition of CBD results in a rearrangement of V649, placing it closer to the central pore axis and further blocking ion permeation. Despite these observations, it is unclear why the authors speculate that this structure was in the "partially open" conformation.

Due to the poor resolution of our detergent structure, we removed it from this version of the manuscript. Text in the manuscript has been revised to reflect these changes.

13) The authors state that CBD-binding leads to opening of the selectivity filter, yet, the largest diameter of the filter is only ~2 Å in diameter, which is insufficient to even allow passage of dehydrated Ca^2+^ ions (diameter = 2.28 Å). Furthermore, it has been speculated for other TRPV family members that ions pass through the selectivity filter in a partially dehydrated form, which would require a pore radius of 1.5-3.5 Å.

Due to the poor resolution of our detergent structure, we removed it from this version of the manuscript. Text in the manuscript has been revised to reflect these changes.

14) The authors make a comparison between the Y575-M672 stabilizing interactions in their CBD-nanodisc structure to the recently-published TRPV3 structures. As the observation of the π-helix in S4-S5 is different between the two structures, i.e., present in the apo TRPV3 structure but missing in the apo TRPV2 structure, this "stabilizing" interaction is not 100% conserved. I encourage the authors to further develop this section of the manuscript to better clarify their message.

Based on the newly obtained TRPV2 channel structures in nanodiscs, we elected not to discuss this in this version of the manuscript.

[Editors' note: further revisions were requested prior to acceptance, as described below.]

Essential revisions:The reviewers feel that the current version of the manuscript is much improved and the authors have made a good effort to address the concerns expressed in the previous revision. However, the reviewers also think that the electrophysiological data is still lacking and does not strongly support the identification of the binding site. In particular, having a similar current magnitude in response to 2-APB between WT and mutant does not constitute evidence that the mutations do not affect sensitivity to 2-APB, because the current magnitude does not distinguish between changes in open probability or level of channel expression. This experiment should therefore be removed from the manuscript.

We agree with the reviewer that similar amplitude responses to 2-APB do not differentiate between open probability and expression level. As requested, we removed this panel from the supplemental figure in the revised version and no longer reference it in the text.

The effect of the double mutation on the CBD to 2-APB current ratio is negligible, despite it being statistically significant. This data can be included in the manuscript, but the conclusion that this mutation identifies the binding site for CBD should be toned down.

We toned down the discussion on the interpretation of the mutant data.